# Patch-Based Composite Adversarial Training against Physically Realizable Attacks on Object Detection

## Abstract

Object detection plays a crucial role in many security-sensitive applications, such as autonomous driving and video surveillance. However, several recent studies have shown that object detectors can be easily fooled by physically realizable attacks, *e.g.*, adversarial patches and recent adversarial textures, which pose realistic and urgent threats. Adversarial Training (AT) has been recognized as the most effective defense against adversarial attacks. While AT has been extensively studied in the $l_\infty$-bounded attack settings on classification models, AT against physically realizable attacks on object detectors has received limited exploration. Early attempts are only performed to defend against adversarial patches, leaving AT against a wider range of physically realizable attacks under-explored. In this work, we consider defending against various physically realizable attacks with a unified AT method. We propose PBCAT, a novel Patch-Based Composite Adversarial Training strategy. PBCAT optimizes the model by incorporating the combination of small-area gradient-guided adversarial patches and imperceptible global adversarial perturbations covering the entire image. With these designs, PBCAT has the potential to defend against not only adversarial patches but also unseen physically realizable attacks such as adversarial textures. Extensive experiments in multiple settings demonstrated that PBCAT significantly improved robustness against various physically realizable attacks over state-of-the-art defense methods. Notably, it improved the detection accuracy by 29.7% over previous defense methods under one recent adversarial texture attack. Code is available at `https://anonymous.4open.science/r/PatchAT`.

## 1 Introduction

Object detection, which requires simultaneously classifying and localizing all objects in an image, is a fundamental task in computer vision. Recent advancements in object detection methods (Zhao et al., 2019; Ren et al., 2017; Tian et al., 2019) have greatly benefited from the utilization of Deep Neural Networks (DNNs). However, DNNs are known to be susceptible to adversarial examples (Szegedy et al., 2014) crafted by adding deliberately designed perturbations to the original examples.

What is worse, adversarial examples exist not only in the digital world but also in the physical world (Thys et al., 2019; Brown et al., 2017; Xu et al., 2020; Wu et al., 2020b; Hu et al., 2022; 2023). Several studies have demonstrated that object detectors can be easily fooled by physically realizable attacks, *e.g.*, adversarial patches (Thys et al., 2019; Brown et al., 2017) and adversarial textures (Hu et al., 2022; 2023). Specifically, adversarial patch attacks craft localized adversarial patterns within a fixed region (*e.g.*, a square patch), while adversarial texture attacks craft more pervasive adversarial perturbations that spread across the entire surface of the object, *e.g.*, adversarial modifications to clothing textures that cover most of the surface of an object. Both adversarial patches and adversarial textures can be implemented in the physical world and are thereby called physically realizable attacks. Given the crucial role of object detection in numerous security-sensitive real-world applications, including autonomous driving (Arnold et al., 2019) and video surveillance (Kumar et al., 2020), it is imperative to improve the adversarial robustness of object detectors against these physically realizable attacks, while poses realistic and severe threats.

Numerous methods have been proposed to defend against adversarial examples, while attackers can still evade most early methods by employing adaptive attacks (Athalye et al., 2018; Tramèr et al., 2020). Among them, Adversarial Training (AT) has been recognized as an effective defense against adaptive adversarial attacks. However, most works (Madry et al., 2018; Zhang et al., 2019a; Li et al., 2023; Zhang & Wang, 2019; Chen et al., 2021; Li et al., 2024b) investigate AT and its variants in the $l_\infty$-bounded attack settings, termed $l_\infty$-*bounded AT* in this work. The $l_\infty$-bounded attacks involve adding a global adversarial perturbation to the images and necessitate manipulation of all image pixels, which are infeasible in the physical world. Thus, the $l_\infty$-bounded AT using such attacks, which are significantly different from physically realizable attacks, cannot defend against physically realizable attacks well (Rao et al., 2020; Wu et al., 2020a; Metzen et al., 2021). Moreover, several works (Rao et al., 2020; Wu et al., 2020a; Metzen et al., 2021; Yu et al., 2021; Zhou et al., 2020; Liu et al., 2022; Kim et al., 2022; Naseer et al., 2019; Yu et al., 2022) have proposed various defense methods against adversarial patches, the simplest form of physically realizable attacks. However, recent adversarial texture attacks (Hu et al., 2022; 2023) create adversarial clothes in the physical world to fool person detectors, which define a different threat model from adversarial patches. To our best knowledge, defense against this form of physically realizable attack has not been explored.

In this work, we consider defending against various physically realizable attacks with a unified AT method. We propose PBCAT, a novel Patch-Based Composite Adversarial Training strategy. We first extend $l_\infty$-bounded AT to AT with adversarial patches, termed *patch-based AT* in this paper, to enhance the robustness against adversarial patch attacks. Secondly, we propose a patch partition and gradient-guided selection method to efficiently find effective patch locations for AT. Thirdly, we incorporate the global imperceptible adversarial perturbations used in $l_\infty$-bounded AT into the patch-based AT which uses small-area gradient-guided adversarial patches. By employing composite perturbations from multiple gradient-guided patches, PBCAT has the potential to defend against not only square adversarial patches but also unseen physically realizable attacks, including adversarial texture attacks with large-area perturbations. Finally, to further enhance the practical utility of PBCAT, we draw inspiration from FreeAT (Shafahi et al., 2019) to enable PBCAT to train a robust object detector at a cost comparable to standard training.

We trained object detectors with PBCAT on the MS-COCO (Lin et al., 2014) dataset. The evaluations were performed on several datasets, including the MS-COCO dataset for the general object detection task as well as the Inria (Dalal & Triggs, 2005) dataset for the downstream security-critical person detection task. We demonstrated that PBCAT significantly improved robustness against various physically realizable attacks over state-of-the-art (SOTA) defense methods in strong adaptive settings (Athalye et al., 2018; Tramèr et al., 2020). On the person detection task, PBCAT secured a Faster R-CNN (Ren et al., 2017) with 60.2% and 56.4% average precision (AP) against two recent adversarial texture attacks, AdvTexture (Hu et al., 2022) and AdvCaT (Hu et al., 2023), respectively. Notably, PBCAT achieved a 29.7% AP improvement over the SOTA defense methods against AdvTexture.

The main contributions of this work can be summarized as follows:

- We propose PBCAT, a novel adversarial training method to defend against various physically realizable attacks with a unified model;

- PBCAT closes the gap between adversarial patches and adversarial textures by patch partition and gradient-guided selection techniques;

- Experiments show that PBCAT achieved promising adversarial robustness over diverse physically realizable attacks in strong adaptive settings.

## 2 PRELIMINARY AND RELATED WORK

### 2.1 ADVERSARIAL ROBUSTNESS ON CLASSIFICATION

Adversarial examples, first discovered on image classification (Szegedy et al., 2014), are input images with deliberately designed perturbations that can fool DNN-based image classifiers while still being easily recognized by humans. Given an image-label pair $(\mathbf{x}, y)$ and a classifier $f_\theta(\cdot)$, the adversarial perturbation $\delta$ can be easily found by maximizing the output loss: $\delta = \arg\max_{\mathcal{B}(\delta) \leq \epsilon} \mathcal{L}(f_\theta(\mathbf{x} + \delta), y)$, where $\mathcal{L}$ denotes a cross-entropy loss, and the attack inten-

sity $\epsilon$ bounds the attack budget $B$. Several approximate methods (Madry et al., 2018; Goodfellow et al., 2015; Carlini & Wagner, 2017) have been proposed to solve the intractable maximizing problem. Projected Gradient Descent (PGD) (Madry et al., 2018) is one of the most popular methods, which optimizes perturbations through multiple iterations with small step sizes. AT and its variants are generally considered the most effective defense methods against adversarial examples, which improve adversarial robustness by incorporating adversarial examples into training:

$$\theta = \arg\min_{\theta} \mathbb{E}_{\mathbf{x}} \{ \max_{\mathcal{B}(\delta) \leq \epsilon} \mathcal{L}(f_{\theta}(\mathbf{x} + \delta), y) \}. \tag{1}$$

However, most works investigate AT on classification models in the $l_{\infty}$-bounded settings (Madry et al., 2018; Zhang et al., 2019a; Li et al., 2024b), *i.e.*, $\mathcal{B}(\cdot) := \| \cdot \|_{\infty}$, which involve adding a global adversarial perturbation to the images and are generally considered as physically infeasible attacks.

## 2.2 ADVERSARIAL ATTACKS ON OBJECT DETECTION

Object detection requires simultaneously classifying and localizing all objects in an image. Modern object detectors (Zhao et al., 2019; Ren et al., 2017; Tian et al., 2019) have significantly improved with the utilization of DNNs. Similar to image classifiers, object detectors are vulnerable to adversarial examples, and several works (Dong et al., 2022; Croce & Hein, 2020; Chen et al., 2021) have investigated how to attack object detectors from various aspects. Unlike classifiers where $l_{\infty}$ attacks and defenses are often investigated, more urgent physically realizable attacks are widely studied for this task, considering that object detection has been widely used in many security-critical applications. Particularly, for the person detection task, several physically realizable adversarial attacks have been proposed. Thys et al. (2019) first propose to generate physically realizable adversarial patches to fool person detectors, which we denote AdvPatch. The AdvTexture attack (Hu et al., 2022) further extends AdvPatch to tileable adversarial textures, offering attack effects from various viewing angles. It proposes a scalable generative method to craft adversarial texture with repetitive structures. The AdvCaT attack (Hu et al., 2023) optimizes adversarial textures into typical camouflage patterns to resemble cloth patterns in the physical world. Attackers can print the clothing texture created by AdvCaT and AdvTexture on a piece of cloth and tailor it into an outfit. Wearing such an outfit can hide the person from SOTA detectors, posing realistic and urgent security threats.

## 2.3 ADVERSARIAL DEFENSES AGAINST PATCH-BASED ATTACKS

To defend against adversarial examples of object detectors, especially patched-based attacks, several types of methods have been proposed. Input preprocessing-based methods, such as LGS (Naseer et al., 2019), SAC (Liu et al., 2022), EPGF (Zhou et al., 2020), and Jedi (Tarchoun et al., 2023), mask out or suppress the potential adversarial patch areas before sending them to the model. Outlier feature filter-based methods, such as FNC (Yu et al., 2021) and APE (Kim et al., 2022), incorporate filters into the model to smooth abnormal inner features caused by adversarial patches. Defensive frame methods, such as UDF (Yu et al., 2022), train an adversarial defense frame surrounding images to improve robustness. However, similar to the experiences on $l_{\infty}$ adversarial defense (Athalye et al., 2018; Tramèr et al., 2020), These non-AT methods can be vulnerable to strong adaptive attacks, as demonstrated in our experimental results in Section 4.2. Besides these empirical methods, some studies (Cohen et al., 2019; Chiang et al., 2020) investigate to improve certified robustness, but till now the certified methods only work under quite tiny perturbations and need time-consuming inference. In this work, we mainly compare PBCAT with empirical methods in practical scenarios.

To the best of our knowledge, only few early works (Wu et al., 2020a; Rao et al., 2020; Metzen et al., 2021) investigate patch-based AT against adversarial patches. Both Rao et al. (2020) and Wu et al. (2020a) examine methods for identifying optimal patch locations for patch-based AT in classification models. Metzen et al. (2021) proposes an enhanced patch-based AT method utilizing meta-learning. Although these patch-based AT methods demonstrate promising results against patch attacks, they are not originally designed for object detection, and adapting them to such task presents significant challenges. For example, both Rao et al. (2020) and Wu et al. (2020a) primarily focus on selecting a single optimal candidate patch position for classification (typically involving one object), whereas object detection requires consideration of numerous bounding boxes, necessitating the identification of multiple patch locations. Consequently, directly applying their proposed methods may lead to an exponential increase in complexity. Furthermore, simply employing adversarial patches for training does not generalize well to a wider range of physically realizable attacks, as discussed later.

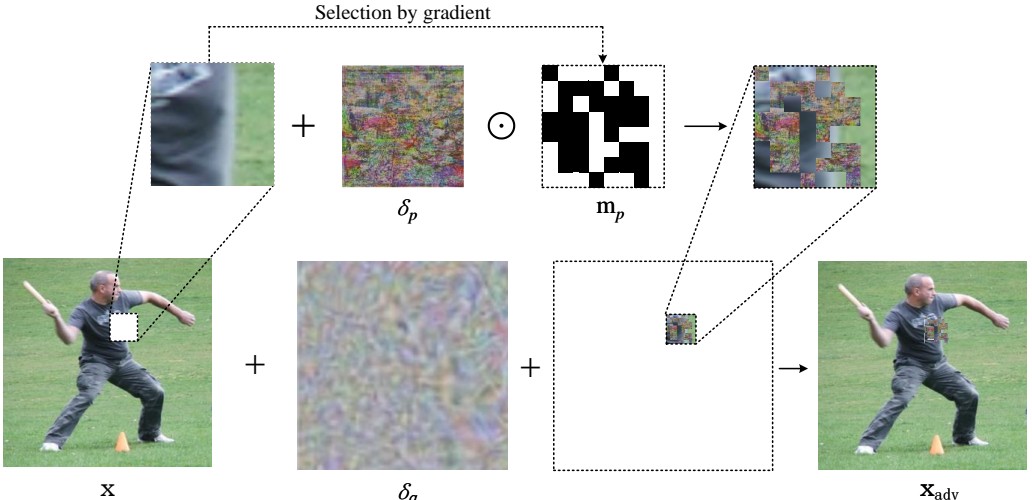

Figure 1: The illustration on how to generate training images for PBCAT. A patch location is randomly selected from each bounding box area first. The patch is then partitioned into multiple small patches. The average gradient norms for each partitioned patch are calculated, and the patches with the top half of the values are selected to obtain a binary mask $\mathbf{m}_p$. Note that $\mathbf{m}_p$ and $\delta_p$ are the same size as the image, but only the local patch area is exhibited here as the surrounding of $\mathbf{m}_p$ is zero. The actual local patch areas used in training are obtained by multiplying $\mathbf{m}_p$ with the adversarial patch $\delta_p$, forming several sub-patches. The global small perturbation $\delta_g$ and these sub-patches are added to obtain the final adversarial example $\mathbf{x}_{\text{adv}}$. Considering that $\delta_g$ is bounded by $\|\delta_g\|_\infty \leq 4/255$, here we scaled $\delta_g$ for better visualization.

## 3 PBCAT

PBCAT aims to defend against various physically realizable attacks by incorporating a combination of a small area of sub-patches and a large area of imperceptible adversarial perturbations. We discuss how to build patch-based AT methods from $l_\infty$-bounded AT in Section 3.1. We then describe the gradient-guided patch partition and selection method in Section 3.2. In Section 3.3, we introduce how to extend patch-based AT to defend against large-area perturbations. Finally, we show how to accelerate PBCAT in Section 3.4.

### 3.1 FROM $l_\infty$-BOUNDED AT TO PATCH-BASED AT

Given an image $\mathbf{x} \in [0, 1]^{3 \times H \times W}$ and its corresponding bounding box labels $y$, where $H \times W$ denotes the input resolution, in the $l_\infty$-bounded attacks, the attackers typically assumes the attack budget $\mathcal{B}$ to be restricted by a bound $\epsilon$: $\|\cdot\|_\infty \leq \epsilon$. $\epsilon$ is often set to a small value in the $l_\infty$ setting, *e.g.*, $4/255$ (Li et al., 2023; Salman et al., 2020). However, $l_\infty$-bounded perturbations constitute *global* perturbations for all pixels in $\mathbf{x}$, which is generally considered physically unrealistic. Physically realizable attacks require restricting the perturbation area to local regions, typically the foreground of an object (Thys et al., 2019; Hu et al., 2022; 2023). In the local regions, the pixels can be perturbed with a large patch perturbation intensity. Different from classification tasks, the input image in object detection usually contains multiple foreground objects. In this work, we assume the threat model that each bounding box may contain an adversarial patch. $l_\infty$-bounded attacks and physically realizable attacks on object detection assume different attack budgets. Considering that AT has sub-optimal effectiveness against unseen threats (Zhang et al., 2019b; Laidlaw et al., 2021), we choose to use patch-based AT to defend against physically realizable attacks.

On the other hand, according to Eq. (1), patch-based AT has a similar formulation with $l_\infty$-bounded AT, except that a mask is needed to restrict the perturbation area. Particularly, the patch-based AT on object detectors can be formulated as below:

$$\theta = \arg\min_\theta \mathbb{E}_\mathbf{x}\{\max_{\|\delta_p \odot \mathbf{m}_p\|_\infty \leq \beta} \mathcal{L}_d(f_\theta(\mathbf{x} + \delta_p \odot \mathbf{m}_p), y)\}, \tag{2}$$

where $\delta_p$ denotes the patch perturbation, while $\mathbf{m}_p$ denotes a binary mask for restricting $\delta_p$ to a local area. $\beta$ denotes a large patch perturbation intensity (*e.g.*, $64/255$ and $1$) and $\mathcal{L}_d$ denotes the detection loss of an object detector that is generally the sum of classification loss and regression loss. The inner maximizing problem of patch-based AT can take inspiration from $l_\infty$-bounded AT, as detailed in Section 3.4. We discuss how to obtain $\mathbf{m}_p$ next.

### 3.2 Gradient-Guided Adversarial Patch Partition and Selection

We specifically discuss $\mathbf{m}_p$ in the case of a single bounding box in an image. For the multiple bounding boxes scenario, $\mathbf{m}_p$ contains multiple patch areas, each of which is similar to the single bounding box case. In the patch attacks (Thys et al., 2019) against object detection, the adversarial patch is often created at a fixed location relative to the bounding box (*e.g.*, the center of the bounding box). However, this may cause information leakage for bounding box prediction, as models might utilize the adversarial patch to perform bounding box regression. PBCAT randomly samples adversarial patch locations within a bounding box to mitigate this issue. Denoting the size of the bounding box as $(w_{\mathrm{bbox}}, h_{\mathrm{bbox}})$, and the center of the bounding box as $(x_{\mathrm{bbox}}, y_{\mathrm{bbox}})$, The center $(x_{\mathrm{s}}, y_{\mathrm{s}})$ of the patch is obtained by randomly sampling patch locations within the bounding box via a Gaussian distribution with $(x_{\mathrm{bbox}}, y_{\mathrm{bbox}})$ as the mean. The width and height of the sampled patch are set to be $s = \lambda \cdot \sqrt{w_{\mathrm{bbox}}^2 + h_{\mathrm{bbox}}^2}$, where $\lambda$ is a hyper-parameter.

The mask $\mathbf{m}_p$ now contains only a large square patch area. As discussed in the early patch-based AT attempts (Rao et al., 2020; Wu et al., 2020a), the patch location plays an important role in enhancing the effectiveness of patch-based AT. However, searching for appropriate locations usually requires several trials in these works, which is time-consuming. For example, to find an appropriate patch location, Rao et al. (2020) requires multiple forward inferences with different patch locations within a specific optimization step, with computational cost proportional to the number of candidate positions. In contrast, we propose to select sub-patches via the gradient information within the sampled large square area, avoiding the time-consuming trials. We first partition the patch into $n \times n$ sub-patches, resulting in $N = n^2$ partitioned areas, where $N$ is an hyper-parameter. Then the average gradient $l_2$ norms for each partitioned area are calculated, and the areas with the top half of values are selected, as the areas with large gradient norms generally are the vulnerable areas that have a significant impact on the output loss. The final mask $\mathbf{m}_p$ used in PBCAT is the mask for the top half of the selected areas. An illustration of $\mathbf{m}_p$ is shown at the top of Fig. 1. Our method ensures to find vulnerable areas for effective AT using a single forward and backward pass, and thus only negligible post-processing computational cost is added compared with previous methods.

### 3.3 Local Patches and Global Noises

Recent adversarial texture attacks (Hu et al., 2022; 2023) adopt a significantly larger area attack than patch-based attack (Thys et al., 2019). Training with small adversarial patches only makes it challenging to defend against these large-area physically realizable attacks. To defend against these attacks, a direct and ideal method would be to perform AT with large-area unrestricted adversarial noises. However, we find that simply increasing the patch size (perturbation area) can induce training collapse of patch-based AT, *i.e.*, the large-area unrestricted perturbation incurred slow training convergence and poor robustness (see Section 4.3.3 and Table 4, where doubling the patch size significantly reduced robustness against various attacks). We guess that the collapse might be caused by the model's limited capacity and significant corruption of object information (a large-area unrestricted patch perturbation can corrupt the information of the entire object).

Instead, we propose to incorporate global imperceptible adversarial perturbations generally used in $l_\infty$-bounded AT into the patch-based AT. The insight behind this approach is as follows: 1) By incorporating large-area adversarial noises while constraining the attack intensity ($l_\infty$ bound), sufficient object information can be kept to avoid training collapse. 2) On the other hand, since the perturbation areas of unseen physically realizable attacks may be located at arbitrary locations of an image (*e.g.*, using a printable patch), the $l_\infty$-bounded global noise ensures the entire image to be covered by adversarial perturbations during training. 3) Training with $l_\infty$-bounded noise has been shown to be helpful against several different adversarial threats (Wang et al., 2024; Li et al., 2024a), and our subsequent results further confirms that $l_\infty$-bounded AT can enhance robustness against

---

**Algorithm 1** "Free" PBCAT on object detection

---

**Require:** Dataset $\mathcal{D}$, $l_\infty$-bounded global perturbation intensity $\epsilon$, patch perturbation step size $\alpha$, patch perturbation intensity $\beta$, replay parameter $r$, model parameters $\theta$, epoch $N_{\mathrm{ep}}$
1: Initialize $\theta$
2: $\delta_g \leftarrow \mathbf{0}; \delta_p \leftarrow \mathbf{0}; \delta \leftarrow \mathbf{0}$
3: **for** epoch $= 1, \ldots, N_{\mathrm{ep}}/r$ **do**
4:     **for** minibatch $B \sim \mathcal{D}$ **do**
5:         **for** i $= 1, \ldots, m$ **do**
6:             Compute gradient of loss with respect to $\mathbf{x}$
7:                 $\mathbf{g}_{\mathrm{adv}} \leftarrow \mathbb{E}_{\mathbf{x} \in B}[\nabla_{\mathbf{x}} \mathcal{L}_d(f_\theta(\mathbf{x} + \delta), y)]$
8:             Update the model parameter
9:                 $\mathbf{g}_\theta \leftarrow \mathbb{E}_{\mathbf{x} \in B}[\nabla_\theta \mathcal{L}_d(f_\theta(\mathbf{x} + \delta), y)]$
10:            update $\theta$ with $\mathbf{g}_\theta$ and the optimizer
11:            Update the patch and global perturbation
12:                 $\delta_g \leftarrow \delta_g + \epsilon \cdot \mathrm{sign}(\mathbf{g}_{\mathrm{adv}})$
13:                 $\delta_p \leftarrow \delta_p + \alpha \cdot \mathrm{sign}(\mathbf{g}_{\mathrm{adv}})$
14:                 $\delta_g \leftarrow \mathrm{clip}(\delta_g, -\epsilon, \epsilon)$
15:                 $\delta_p \leftarrow \mathrm{clip}(\delta_p, -\beta, \beta)$
16:            Generate mask $\mathbf{m}_p$ by the steps in Section 3.2 and update the final perturbation
17:                 $\delta \leftarrow \delta_p \odot \mathbf{m}_p + \delta_g$
18:         **end for**
19:     **end for**
20: **end for**

---

patch attacks. The final perturbation used in PBCAT is:

$$\delta = \delta_p \odot \mathbf{m}_p + \delta_g, \tag{3}$$

where $\delta_g$ denotes the global noises, *i.e.*, $\|\delta_g\|_\infty \leq \epsilon$.

### 3.4 ACCELERATING AT

Early patch-based AT works (Wu et al., 2020a; Rao et al., 2020; Metzen et al., 2021) use the full PGD attack to perform the inner maximizing problem of AT and train the object detectors from scratch, which is quite time-consuming. In PBCAT, taking inspiration from recent SOTA $l_\infty$-bounded AT practice (Li et al., 2023) on object detection, we opt for FreeAT (Shafahi et al., 2019) as the default setting for patch-based AT on object detection and use adversarially pre-trained backbone network (Li et al., 2023). FreeAT recycles gradient perturbations to reduce the extra training costs brought by inner maximizing while achieving comparable adversarial robustness. We first initialize the patch $\delta_p$ and the global perturbation $\delta_g$ to zero. At each iteration, we calculate the gradient and take its signs, multiplying it by different step sizes to update $\delta_p$ and $\delta_g$, respectively. The pseudo-code of "Free" PBCAT on object detection is provided in Algorithm 1. With "Free" PBCAT, the training cost of AT can be reduced to be comparable to the standard training. The actual training time is shown in Appendix A.

## 4 EXPERIMENTS

### 4.1 EXPERIMENTAL SETTINGS

Unless otherwise specified, we performed experiments on the popular two-stage detector Faster R-CNN (Ren et al., 2017) with a ResNet-50 (He et al., 2016) as the backbone. More detectors were evaluated in Section 4.4. We trained the general object detector with PBCAT. But considering that most physically realizable attacks (Thys et al., 2019; Brown et al., 2017; Hu et al., 2022; 2023) were proposed on the security-critical downstream tasks, *e.g.*, the person detection task, we evaluated the detector trained with PBCAT on the person detection task, too.

**Datasets and metrics.** Three datasets are used in this work. 1) The MS-COCO (Lin et al., 2014) dataset for training the general object detectors. We used the 2017 version, containing 118,287

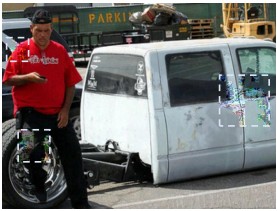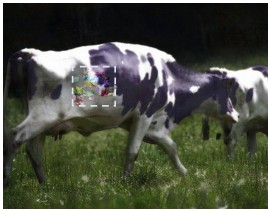

Figure 2: Visualization of training examples with adversarial perturbations used in PBCAT. Perceptible gradient-guided adversarial patches and imperceptible global adversarial perturbations are added to each image. The patch regions are annotated with white dashed boxes in the figure.

images of 80 object categories for training and 5,000 images for evaluation. 2) The Inria Person (Dalal & Triggs, 2005) dataset for the person detection task. Note that this dataset is only used for attack evaluation. 3) The synthetic dataset used in Hu et al. (2022; 2023) for evaluation. This dataset contains 506 background images of different scenes, with 376 images used for adversarial texture optimization and 130 images for attack evaluation. We rendered a 3D person wearing an adversarial outfit optimized by AdvTexture and AdvCaT on the provided background images using a differential renderer (Ravi et al., 2020). We used $AP_{50}$, the Average Precision with an IoU threshold of 0.5, as the primary metric for evaluating the detection accuracy under different attacks, considering that it is a widely-used and practical metric for object detection (Li et al., 2023; Redmon & Farhadi, 2018).

**Training recipe of PBCAT.** We trained the object detector with PBCAT on the MS-COCO dataset. Unless otherwise specified, the patches in each image were generated with a patch perturbation step size $\alpha = 8/255$, patch perturbation intensity $\beta = 64/255$, scale factor $\lambda = \sqrt{2}/5$, and the amount of sub-patches $N = 64$. The replay parameter for FreeAT was set to be $r = 8$. Specifically, each bounding box had a 50% chance of being attached to an adversarial patch to increase the detection accuracy for objects without adversarial patches (clean objects). The global perturbations were generated with a perturbation intensity $\epsilon = 4/255$. Additional training settings basically followed the recipe proposed by Li et al. (2023) (see Appendix A), which resulted in the recent SOTA robustness against the $l_\infty$-bounded attacks. We show some training examples with adversarial perturbations used in PBCAT in Fig. 2.

**Attack evaluation setups.** For the general detection task on MS-COCO, our attack evaluation used the masked PGD attack to create the adversarial square patch for each bounding box, termed PGDPatch attack. Here the iteration step was set to 200, the step size was set to 2, and the hyper-parameter for the patch size was set to $\lambda = 1/5\sqrt{2}$, resulting in a patch area of 1% to 5% relative to the area of the bounding box. Note that the physical implementation of the patches created by PGDPatch was relatively difficult because the tricks for physical implementation like the TV loss (Sharif et al., 2016) were not used. Instead, we cared more about the security-critical person detection task, where three actual physically realizable attacks were evaluated: AdvPatch (Thys et al., 2019), AdvTexture (Hu et al., 2022), and AdvCaT (Hu et al., 2023). The evaluation settings for these three attacks strictly followed their original configurations in the digital world: The detector trained on MS-COCO was evaluated on the downstream person detection task directly. All of the three attacks also have their implementations and evaluations in the physical world. However, validating the effectiveness of defense methods against these physically realizable attacks in the digital world is sufficient, because the attack success rates of physically realizable attacks in the digital world is typically higher than those in the real physical world (Thys et al., 2019; Hu et al., 2022). In real physical world, physical implementation errors and differences in physical conditions (*e.g.*, illumination) decrease the attack success rates. Thus, if a defense method performs well in defending against these physically realizable attacks in the digital world, it can exhibit stronger robustness in the real physical world. Similar evaluation paradigms have also been adopted by many defense methods such as Wu et al. (2020a); Zhou et al. (2020); Liu et al. (2022).

## 4.2 ROBUSTNESS AGAINST ADVERSARIAL ATTACKS

We first compared PBCAT with the recent SOTA AT method (Li et al., 2023) against the $l_\infty$-bounded attacks, whose training recipe was also adopted by ours, on the general object detection task. The

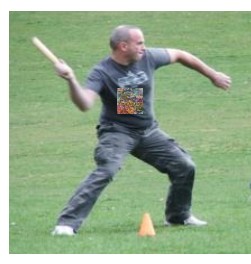 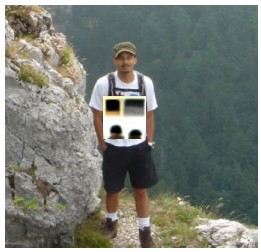 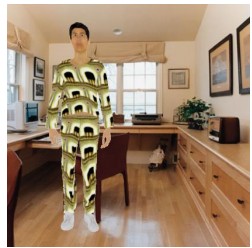 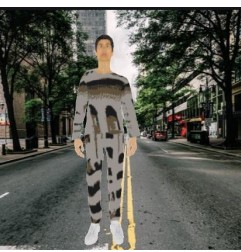

| PGDPatch | AdvPatch | AdvTexture | AdvCaT |

Figure 3: Visualization of examples of four attacks used in our evaluation. PGDPatch was optimized on MS-COCO. AdvPatch was optimized on the Inria dataset. AdvTexture and AdvCaT were optimized on the synthetic dataset used in Hu et al. (2023). PGDPatch is used to evaluate the general object detection task while other attacks were used to evaluate the downstream person detection.

Table 1: The detection accuracies ($AP_{50}$) of models with different defense methods under adaptive attacks. Clean (Inria) and AdvPatch (Thys et al., 2019) were evaluated on the Inria dataset. Clean (Synthetic), AdvTexture (Hu et al., 2022), and AdvCaT (Hu et al., 2023) were evaluated on the synthetic dataset.

| Method | Clean (Inria) | AdvPatch | Clean (Synthetic) | AdvTexture | AdvCaT |
|---|---|---|---|---|---|
| Vanilla | 96.2 | 37.3 | 86.4 | 0.2 | 0.4 |
| LGS (Naseer et al., 2019) | 95.9 | 24.1 | 86.4 | 6.2 | 4.3 |
| SAC (Liu et al., 2022) | 94.8 | 57.1 | 85.4 | 0.7 | 0.6 |
| EPGF (Zhou et al., 2020) | 95.1 | 43.2 | 86.7 | 2.9 | 0.4 |
| Jedi (Tarchoun et al., 2023) | 90.2 | 32.8 | 88.1 | 3.8 | 2.3 |
| FNC (Yu et al., 2021) | 96.8 | 53.0 | 81.9 | 6.0 | 5.8 |
| APE (Kim et al., 2022) | 95.3 | 46.7 | 81.9 | 0.0 | 0.4 |
| UDF (Yu et al., 2022) | 69.1 | 19.3 | 84.9 | 2.2 | 5.8 |
| PatchZero (Xu et al., 2023) | 96.2 | 38.5 | 79.4 | 0.0 | 0.2 |
| NAPGuard (Wu et al., 2024) | 96.1 | 47.0 | 81.1 | 2.2 | 0.4 |
| $l_\infty$-Bounded AT (Li et al., 2023) | 95.9 | 56.1 | 92.5 | 30.5 | 39.6 |
| PBCAT (Ours) | 95.4 | **77.6** | 92.5 | **60.2** | **56.4** |

results detailed in Appendix B show that PBCAT achieved an $AP_{50}$ of $37.8\%$ averaged on each object category and an $AP_{50}$ of $34.5\%$ on the particular person category under PGDPatch attack, surpassing Li et al. (2023) by $6.1\%$ and $4.4\%$, respectively. These results show the effectiveness of PBCAT against potential physically realistic attacks on the general object detection task. We then turn to the actual physically realistic attacks on the security-critical person detection task.

In the person detection task, we compared PBCAT with various defense approaches against patch-based attacks, including input preprocessing-based methods, outlier feature filter-based methods, and defensive frame methods (see Section 2.3). These defense methods were applied to the standardly trained detector, denoted as *Vanilla*. We also compared PBCAT with the SOTA $l_\infty$-bounded AT method (Li et al., 2023). Please note that early patch-based AT works (Wu et al., 2020a; Rao et al., 2020; Metzen et al., 2021) were not evaluated as these methods were proposed originally for the classification task and it is challenging to adapt to object detection (see Section 2.3). All of these defense methods were evaluated in the white-box adaptive setting to show the worst-case robustness.

The results of different defense methods are shown in Table 1. These non-AT defense methods are all vulnerable to the adaptive patch attack. For the AdvPatch attack, the best non-AT defense method was SAC (Liu et al., 2022), achieving $57.1\%$ $AP_{50}$. Moreover, against the stronger adaptive adversarial texture attacks, all these non-AT defense methods were broken. Interestingly, we found that the $l_\infty$-bounded AT (Li et al., 2023)) outperformed all non-AT defense methods. We note that Li et al. (2023) did not evaluate the models trained with $l_\infty$-bounded AT against physically realizable attacks in their original work, as it was originally for defending against $l_\infty$-bounded attacks. Nevertheless, our results show that $l_\infty$-bounded AT enhanced robustness against physically realizable attacks as well. Compared with $l_\infty$-bounded AT, PBCAT further improved the robustness. Notably, against

the AdvTexture attack, PBCAT improved the detection accuracy by 29.7%. Appendix C visualizes some detection results of the model trained with PBCAT under physically realizable attacks.

Despite the distinct features between training examples (Fig. 2) and the evaluated attack examples (Fig. 3), PBCAT model had strong robustness against various attacks, showing its good transferability across different attacks. To further validate this, we evaluated two transfer-based patch attacks (Huang et al., 2023; Hu et al., 2021) on our model, and the results detailed in Appendix D show that these transfer attacks almost lose the ability to fool the detectors trained with PBCAT.

### 4.3 ABLATION STUDY

#### 4.3.1 EFFECTIVENESS OF EACH DESIGN IN PBCAT

Table 2: The detection accuracies ($AP_{50}$) of models trained with different ablation settings. Clean (COCO) and PGDPatch were evaluated on MS-COCO; AdvPatch (Thys et al., 2019) was evaluated on Inria; AdvTexture (Hu et al., 2022) and AdvCaT (Hu et al., 2023) were evaluated on the synthetic dataset used in Hu et al. (2023).

| Patch | Global | Partition | Gradient | Clean (COCO) | PGDPatch | AdvPatch | AdvTexture | AdvCaT |
|:---:|:---:|:---:|:---:|:---:|:---:|:---:|:---:|:---:|
| ✓ | | | | 54.4 | 29.5 | 35.4 | 1.6 | 0.8 |
| | ✓ | | | 51.2 | 30.7 | 56.1 | 30.5 | 39.6 |
| ✓ | ✓ | | | 45.3 | 21.2 | 72.8 | 24.9 | 19.5 |
| ✓ | ✓ | ✓ | | 45.9 | 30.6 | 59.0 | 14.2 | 47.0 |
| ✓ | ✓ | ✓ | ✓ | 45.6 | **37.8** | **77.6** | **63.3** | **56.4** |

We first conducted an ablation study to show the effectiveness of each design in PBCAT: using the small area patch perturbations, using the global imperceptible noise perturbations, using the patch partition strategy, and using the gradient-guided selection technique, denoted as "Patch", "Global", "Partition", and "Gradient", respectively. When "Partition" is used and "Gradient" is absent, we employ the same patch partitioning method to divide a sampled patch into sub-patches, while the selection of the sub-patches is done randomly. The results are shown in Table 2. We can see that all of these designs contribute to enhancing robustness against physically realizable attacks. By incorporating the global perturbations in conjunction with adversarial patches, the robust detection accuracy against the AdvPatch attack increased to 72.8%. Introducing the patch partitioning strategy and the gradient-guided selection technique further improved robustness across all attacks. Additionally, the results were sub-optimal when the patches were partitioned, but the retained sub-patches were selected randomly (the second-to-last row of Table 2), instead of by gradient guidance.

#### 4.3.2 THE NUMBER OF SUB-PATCHES

Since our sampled large patch is partitioned into sub-patches, an exploration is required to identify the optimal amount of partitioning. We conducted three experiments, varying the number of sub-patches $N$: 16 ($4 \times 4$), 64 ($8 \times 8$), and each pixel within the patch as a sub-patch (pixel-level). Other settings in this experiment followed Section 4.1. The results shown in Table 3 indicate that a balance was required in the amount of sub-patches.

#### 4.3.3 THE SIZE OF THE SAMPLED PATCH

The hyper-parameter $\lambda$ controlled the size of the sampled patch, representing the ratio of the edge length of a square patch to the diagonal length of the bounding box. We consider three models

Table 3: The detection accuracies ($AP_{50}$) of models trained with different numbers of sub-patches. Pixel-level indicates that each pixel is considered as a sub-patch.

| Sub-patches | AdvPatch | AdvTexture | AdvCaT |
|:---:|:---:|:---:|:---:|
| 16 | **78.3** | 50.8 | 46.2 |
| 64 | 77.6 | **60.2** | 56.4 |
| Pixel-level | 67.4 | 20.4 | **59.4** |

Table 4: The detection accuracies ($AP_{50}$) of models trained with different sizes of sampled patches. Here the number of the sub-patches was set to 16.

| Scale factor $\lambda$ | AdvPatch | AdvTexture | AdvCaT |
|:---:|:---:|:---:|:---:|
| $2\sqrt{2}/10$ | 78.3 | **50.8** | **46.2** |
| $3\sqrt{2}/10$ | **80.4** | 49.1 | 38.5 |
| $4\sqrt{2}/10$ | 63.4 | 24.6 | 43.6 |

Table 5: The detection accuracies ($AP_{50}$) of the FCOS models trained with different methods against attacks on different tasks. The same conventions are used as in Table 2.

| Method | Clean (COCO) | PGDPatch | AdvPatch | AdvTexture | AdvCaT |
|---|---|---|---|---|---|
| Vanilla | 56.0 | 17.6 | 28.3 | 0.0 | 0.1 |
| $l_\infty$-Bounded AT | 49.5 | 26.7 | 29.9 | 26.7 | 17.7 |
| PBCAT (Ours) | 43.7 | **27.8** | **58.0** | **55.1** | **26.0** |

Table 6: The detection accuracies ($AP_{50}$) of the DN-DETR models trained with different methods against attacks on different tasks. The same conventions are used as in Table 2.

| Method | Clean (COCO) | PGDPatch | AdvPatch | AdvTexture | AdvCaT |
|---|---|---|---|---|---|
| Vanilla | 58.5 | 7.3 | 2.4 | 0.3 | 2.8 |
| $l_\infty$-Bounded AT | 45.7 | 32.0 | 23.5 | 0.0 | 0.1 |
| PBCAT (Ours) | 45.3 | **32.7** | **56.3** | **16.8** | **56.8** |

with different $\lambda$ values to examine their effect. The results shown in Table 4 indicate that simply enlarging the size of the sampled patch was ineffective when defending against large-area texture attacks. Moreover, too large sampled patches had negative effects even when defending against the patch-based attack.

### 4.4 EFFECTIVENESS ACROSS OBJECT DETECTORS

PBCAT requires no assumption about the structure of the detector and has shown success on the two-stage Faster R-CNN detector in the above experiments. Here we evaluated PBCAT with two additional detectors to validate its effectiveness on more models. Here we used FCOS (Tian et al., 2019), a typical single-stage object detector, and DN-DETR (Li et al., 2022), a transformer-based detector. Note that Faster R-CNN, FCOS, and DN-DETR have distinct structures.

The detectors were trained on MS-COCO. Three methods were evaluated for each detector: standard training, $l_\infty$-bounded AT, and PBCAT. The patch perturbation step size was $\alpha = 4/255$. Patch perturbation intensity was $\beta = 32/255$ and patch size $\lambda$ was 0.2. Other settings basically followed those on Faster R-CNN. The detectors with standard training was taken from the `mmdetection` (Chen et al., 2019) repository directly. The evaluation results on FCOS and DN-DETR are shown in Table 5 and Table 6, respectively. FCOS trained by PBCAT achieved an $AP_{50}$ of 58.0% on the Inria dataset under the AdvPatch attack, surpassing $l_\infty$-bounded AT by 28.1%. Its $AP_{50}$ on AdvTexture and AdvCaT are also much higher than that in Vanilla and $l_\infty$-bounded AT. Similarly, PBCAT significantly improved the robustness of DN-DETR. These results show the effectiveness of PBCAT across diverse object detectors.

## 5 DISCUSSION AND CONCLUSION

In this work, we introduce PBCAT, a novel adversarial training method to defend against physically realizable attacks. With extensive experiments on both the general object detection task and the security-critical person detection task, we demonstrated the effectiveness of PBCAT across various scenarios under strong adaptive attacks. Notably, PBCAT demonstrates significant improvements over previous SOTA $l_\infty$-bounded AT method when defending against the AdvTexture (Hu et al., 2022) attack. We encourage future work in enhancing adversarial robustness to consider a broader range of attacks beyond $l_\infty$-bounded attacks and patch-based attacks. Additionally, our work highlights that AT is still one of the most promising ways to achieve robustness against physically realizable attacks. The social impact of this work is discussed in Appendix E.

**Limitation.** Similar to most AT works (Zhang et al., 2019a; Li et al., 2023), PBCAT slightly decreased the clean accuracies of detectors on the complex MS-COCO dataset. While on the Inria dataset, the object detector trained with PBCAT exhibited strong clean accuracy comparable to the detector with standard training. It is an open question whether there is an internal trade-off between robustness against physically realizable attacks and clean accuracy. We leave it to be future work.

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

## A    ADDITIONAL TRAINING SETTINGS

We used the adversarially pre-trained checkpoint provided by Salman et al. (2020) as the initialization of the backbone and trained the detector on MS-COCO for 48 epochs with the AdamW optimizer and an initial learning rate of 0.0001. For the learning rate schedule, the detector used multi-step decay that scaled the learning rate by 0.1 after the 40th epoch. The input images were resized to have their shorter side being 800 and their longer side less or equal to 1333 during training.

We compared the time cost of PBCAT with standard adversarial training in Table A1. All training was conducted on 8 NVIDIA 3090 GPUs. For Faster R-CNN with PBCAT, the training required about 44 hours, while standard training required about 34 hours under the same conditions. The minor difference can be attributed to the additional cost incurred by the gradient post-processing process (partial partitioning and selection). Therefore, we conclude that PBCAT has a training cost that is comparable to that of standard training.

Table A1: The comparison between training time (in hours) of PBCAT and $l_\infty$-Bounded AT.

| Method | Faster-RCNN | FCOS | DN-DETR |
|---|---|---|---|
| $l_\infty$-Bounded AT (Li et al., 2023) | 34h | 26h | 32h |
| PBCAT (Ours) | 44h | 38h | 48h |

Table A2: The detection accuracies ($AP_{50}$) of models trained with various methods on MS-COCO.

| Method | Clean | | PGDPatch | |
|---|---|---|---|---|
| | All | Person | All | Person |
| Vanilla | 58.1 | 52.0 | 18.4 | 17.5 |
| $l_\infty$-Bounded AT (Li et al., 2023) | 51.2 | 45.3 | 30.7 | 30.1 |
| PBCAT (Ours) | 45.6 | 41.4 | **37.8** | **34.5** |

## B    ADDITIONAL EVALUATION ON MS-COCO

Table A2 shows the results of models trained with various methods on MS-COCO. "All" and "Person" in Table A2 denote the averaged $AP_{50}$ on each object category and the $AP_{50}$ on the particular person category, respectively.

## C    VISUALIZATION RESULTS OF DETECTORS WITH PBCAT

Some visualization results of the Faster R-CNN trained with PBCAT are shown in Fig. A1. We can see that the detector performed quite well under the large-area and strong adversarial texture attacks.

## D    ROBUSTNESS AGAINST TRANSFER ATTACKS

To further validate that the models trained with PBCAT can have strong robustness against more unseen physically realizable attacks, we additionally evaluated two transfer-based patch attacks (Huang et al., 2023; Hu et al., 2021) on the model trained with PBCAT. The adversarial patches were generated based on their original settings on the vanilla (clean) Faster R-CNN model. The patches were applied to the images in the Inria dataset to evaluate the $AP_{50}$ of Faster R-CNN trained by PBCAT. The results are shown in Table A3. We can see that these transfer-based patch attacks almost lose the ability to fool the detectors trained with PBCAT.

Additionally, we used the three types of detectors we trained in this work (as discussed in Section 4.4), Faster R-CNN, FCOS (Tian et al., 2019), DN-DETR (Li et al., 2022), to perform the black-box transfer attacks. Here we used the AdvPatch attack on the Inria dataset. The results are shown in Table A4. We can also observe that the models trained with our PBCAT can defend these black-box transfer-based attacks better than white-box attacks.

Table A3: The detection accuracies ($AP_{50}$) of the Faster R-CNN model on transfer-based patch attacks on the Inria dataset.

| Method | T-SEA (Huang et al., 2023) | NatPatch (Hu et al., 2021) |
|---|---|---|
| Vanilla | 31.7 | 54.4 |
| PBCAT (Ours) | **90.9** | **86.3** |

Table A4: The detection accuracies ($AP_{50}$) under AdvPatch on the Inria dataset in the transfer-based attack setting. The adversarial examples generated on the source models (each column) were fed into the target models (each row).

| Source \ Target | Faster-RCNN | FCOS | DN-DETR |
|---|---|---|---|
| Faster-RCNN | 77.6 | 80.7 | 83.1 |
| FCOS | 80.0 | 58.0 | 79.3 |
| DN-DETR | 69.2 | 59.9 | 56.3 |

## E    BROADER IMPACT

Our method increases the robustness of object detectors against physically realizable attacks. This could potentially lead to more effective surveillance systems, which could encroach upon personal privacy if misused. However, we believe the concrete positive impact on security generally outweighs the potential negative impacts. Robust object detectors can enhance various beneficial applications, such as autonomous driving, video surveillance for public safety, and other critical systems. Nonetheless, it is crucial to develop and deploy such technologies responsibly, with ethical considerations to mitigate potential misuse and protect individual privacy.

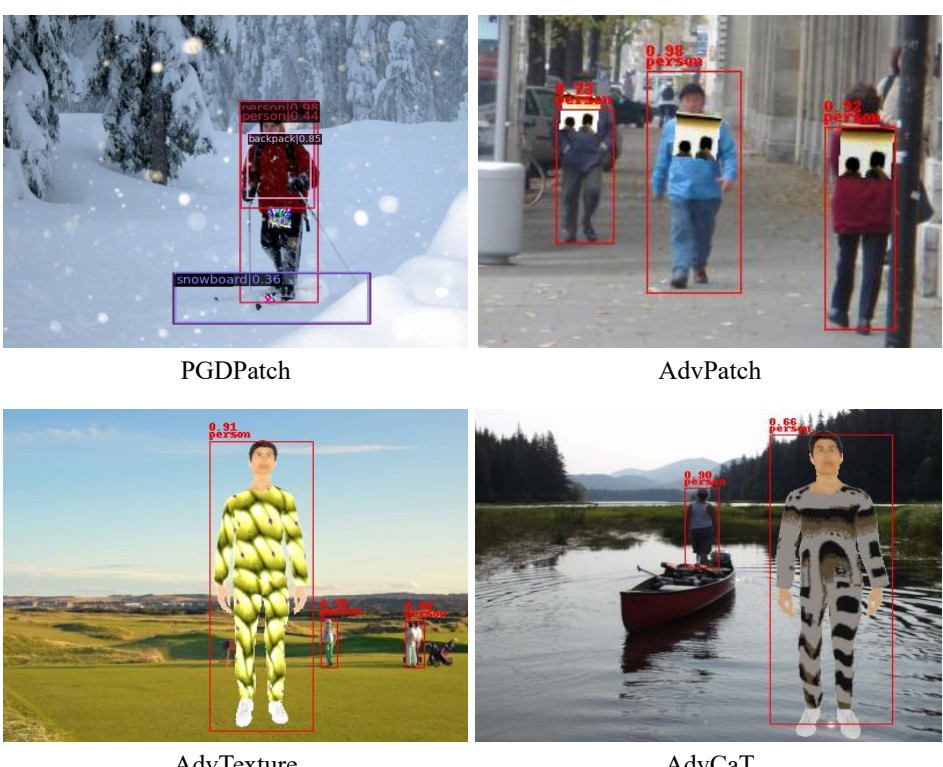

PGDPatch                              AdvPatch

AdvTexture                              AdvCaT

Figure A1: The detection results of the model trained with PBCAT under various physically realizable attacks. The detected bounding boxes with confidence larger than 0.5 are visualized.

