# OpenReview forum: "PBCAT: Patch-Based Composite Adversarial Training against Physically Realizable Attacks on Object Detection"
_ICLR.cc/2025/Conference — Submitted to ICLR 2025_

### Official Review · Reviewer_y8aD · 2024-10-20

**Soundness:** 3
**Presentation:** 3
**Contribution:** 2
**Rating:** 5
**Confidence:** 4

**Summary:**

The authors introduce a patch-based adversarial training technique designed to improve the robustness of object detection models against both patch-based and more recent texture-based attacks. The method involves two types of perturbations: local perturbations applied to the attacked object and a global perturbation affecting the entire image. The global perturbation is aimed at enhancing the robustness against texture-based attacks. In their evaluation, the authors compare their technique to one adversarial training (AT) approach and several non-AT methods across three patch-based attacks. They also present ablation studies to assess the impact of various hyperparameters. Finally, the evaluation is extended to other object detection models to demonstrate the method's broader applicability.

**Strengths:**

1. The paper addresses a practical yet underexplored topic: adversarial training for defending object detection models against realizable attacks.
2. The evaluation setup is well-detailed, and the provided code ensures easy reproducibility.
3. The proposed method achieves excellent performance in terms of adversarial robustness.

**Weaknesses:**

1. Incomplete literature review – while the authors state that there are no previous works that specifically propose patch-based AT for object detection, a more in-depth review of the literature would have revealed that techniques such as Ad-YOLO [1] and PatchZero [2] already exist (and should be compared to). Additionally, including comparisons to more recent non-AT methods (e.g., PatchBreaker [3], NAPGuard [4]) would strengthen the paper's overall contribution.‏

2. Lack of novelty – The proposed method appears relatively simple, primarily combining existing techniques adapted for object detection without introducing substantial new contributions, aside from the patch partitioning and selection strategy.

3. Experiments - While the authors conduct a relatively comprehensive evaluation, several aspects are lacking:

* Models: Since the focus is on person detection, which typically involves real-time scenarios, the evaluation should prioritize low-latency models (e.g., one-stage detectors) rather than slower ones like Faster R-CNN. Including YOLO models, particularly the most recent versions, would have been more relevant, as they are widely used in real-time object detection.
* "Clean" results: While the authors acknowledge the performance drop on clean images as a limitation, the degradation in accuracy is significant, especially when compared to (Li et al. 2023) in Tables A1, 5, and 6. This raises concerns about whether the improved robustness stems from a robustness-accuracy trade-off. A more fair comparison would require matching the AP on clean images across methods before assessing robustness.
* Results discussion: The results are presented with limited interpretation. The discussion would benefit from addressing edge cases and explaining unintuitive findings (as highlighted in question 4 below).

4. Presentation - the submission is held back by the writing quality, particularly in the method section, mainly focused around the partially existing formulaic descriptions. For instance, the number of selected sub-patches should be parametrized (with an accompanying equation or algorithm) to better align with the presentation of the ablation study in Section 4.3.2.

Minor comments:
- Algorithm 1 – the use of $m$ and $m_p$ is confusing.
- The placement of the tables on Page 9 makes them hard to read.
- Best “Clean” performance should also be marked with bold.

[1] Ji, N., Feng, Y., Xie, H., Xiang, X., & Liu, N. (2021). Adversarial yolo: Defense human detection patch attacks via detecting adversarial patches. arXiv preprint arXiv:2103.08860.

[2] Xu, K., Xiao, Y., Zheng, Z., Cai, K., & Nevatia, R. (2023). Patchzero: Defending against adversarial patch attacks by detecting and zeroing the patch. In Proceedings of the IEEE/CVF Winter Conference on Applications of Computer Vision (pp. 4632-4641).

[3] Huang, S., Ye, F., Huang, Z., Li, W., Huang, T., & Huang, L. (2024). PatchBreaker: defending against adversarial attacks by cutting-inpainting patches and joint adversarial training. Applied Intelligence, 54(21), 10819-10832.

[4] Wu, S., Wang, J., Zhao, J., Wang, Y., & Liu, X. (2024). NAPGuard: Towards Detecting Naturalistic Adversarial Patches. In Proceedings of the IEEE/CVF Conference on Computer Vision and Pattern Recognition (pp. 24367-24376).

[5] Liu, X., Yang, H., Liu, Z., Song, L., Li, H., & Chen, Y. (2018). Dpatch: An adversarial patch attack on object detectors. arXiv preprint arXiv:1806.02299.

‏

**Questions:**

1. It is unclear whether the patch selection calculation (i.e., the $l_2$ norm calculations) is performed on the clean image or the adversarial image (the one containing the adversarial patch).
* Could you please clarify this?
* Additionally, what is the rationale behind choosing a square-shaped mask?
* Have you considered experimenting with different norms beyond the $l_2$ norm?

2. In the model training, are the weights initialized to random values or pre-trained weights? If random initialization is used, the object detector may risk overfitting on the Inria dataset, which contains only a few hundred images. This could explain the inconsistencies observed between the results on MS-COCO and Inria.

3. In line 345, the total number of sub-patches is set to $n^2=64$ , and in lines 238-239, you mention that the top half are selected, indicating that 32 patches are chosen. However, in the ablation study regarding the number of sub-patches used during the selection process (Table 3), only a single value (16) is presented as a portion of the sub-patches, since using 64 means utilizing the entire set. This leads me to infer that 16 is deemed the optimal value. Does using 32 sub-patches result in better performance? It would be beneficial to explore additional values in this experiment.

4. Could you provide some insights into the results presented in Table 2, particularly concerning the "Global" component? I find it challenging to understand why the "Global" component enhances robustness against AdvTexture and AdvCat attacks, given the significant differences in perturbation styles between them. Additionally, why does robustness decrease against AdvTexture when the Patch and Partition components are added (Lines 3 and 4)?

5. Following the above question, in line 465 it is stated that “Partition” denotes the patch partition strategy. What is the strategy other than “Gradient”? what does Line 4 in Table 2 mean?

5. While I acknowledge that the paper focuses on patches attached to objects, it would be beneficial to evaluate the proposed approach against attacks that place patches in different locations (e.g., DPatch [5]) and to study the effect of the "Global" component on these attacks. Demonstrating the ability to mitigate the impact of such patches could significantly enhance the paper's contributions.

---

> ### Author Response · Authors · 2024-11-24
> **Thank you for the valuable review (1/4)**
>
> Thank you for the effort of review. We are happy to see that your think that our work addresses a practical yet underexplored topic. **We have uploaded a revised version of our paper, with revisions highlighted in blue for clarity.**  Below we address the detailed comments, and hope that you may find our response satisfactory.
>
> **W1: Incomplete literature review - while the authors state that there are no previous works that specifically propose patch-based AT for object detection, a more in-depth review of the literature would have revealed that techniques such as Ad-YOLO and PatchZero already exist. Additionally, including comparisons to more recent non-AT methods (e.g., PatchBreaker, NAPGuard) would strengthen the paper's overall contribution.**
>
> Thank you for your valuable suggestions. We would like to first clarify that PatchZero [1] is generally not considered as patch-based adversarial training (AT) methods. PatchZero used additional pre-processing detection modules to detect the adversarial patches and then masked the detected areas in an image before sending to the object detectors. It is essentially input preprocessing-based defense methods (see Section 2.3). In contrast, patch-based AT does not need any additional preprocessing modules and directly robustify the object detector itself, which provides the internal robustness of the object detectors. We have compared many input preprocessing-based defense methods, such as LGS, SAC, EPGF, Jedi, etc. The results shown in Table 1 indicates that due to obfuscated gradients [2], most of these preprocessing-based defense method cannot defend against adaptive attacks. But as per your suggestion, we have added the evaluation of PatchZero, as shown below. It cannot defend against strong adversarial texture attacks. In addition, we have also compared our method with the recent NAPGuard [3] method. The results show that our method has significant performance advantage over them. We have added these results to Table 1.
>
> As for Ad-YOLO [4] and PatchBreaker [5], unfortunately, we could not find the open-source code or model checkpoints for this work. We also contacted the authors but received no response before the rebuttal. **Thus we apologize that we cannot provide a meaningful comparison.** But we notice that PatchBreaker has the high similarity with PatchZero (both use a module to detect and remove adversarial patches). Additionally, according to the NAPGuard paper, NAPGuard has significantly better robustness over Ad-YOLO. Thus, we believe that our additional results of NAPGuard and PatchZero can represent the methods you mentioned and PBCAT has obvious advantages over these methods.
>
>
> |||||||
> |-|-|-|-|-|-|
> |Method|Clean(Inria)|Advpatch|Clean(Synthetic)|AdvTexture|AdvCaT|
> |PatchZero|96.2|38.5|79.4|0.0|0.2|
> |NAPGuard|96.1|47.0|81.1|2.2|0.4|
> |PBCAT|92.5|**77.6**|92.5|**60.2**|**56.4**|
>
>
>
>
> [1] Xu, K., et al. Patchzero: Defending against adversarial patch attacks by detecting and zeroing the patch. WACV, 2023.
>
> [2] Athalye A, et al. Obfuscated gradients give a false sense of security: Circumventing defenses to adversarial examples, ICML, 2018.
>
> [3] Wu, S., et al. NAPGuard: Towards Detecting Naturalistic Adversarial Patches. CVPR, 2024.
>
> [4] Ji, N., et al. Adversarial yolo: Defense human detection patch attacks via detecting adversarial patches. arXiv, 2021.
>
> [5] Huang, S, et al. PatchBreaker: defending against adversarial attacks by cutting-inpainting patches and joint adversarial training. Applied Intelligence, 2024.
>
>
>
>
>
> **W2. Lack of novelty - The proposed method appears relatively simple, primarily combining existing techniques adapted for object detection without introducing substantial new contributions, aside from the patch partitioning and selection strategy.**
>
> As an AT method, PBCAT inherits the framework of adversarial training, and thus it may seem familiar at first glance. However, as an effective method defending against physically realizable attacks, PBCAT can improve the robustness of different object detectors by a large margin (see Table1). These improvements can be attribute to the novel and well-motivated design of PBCAT: the patch partitioning and selection strategy, which is distinct from previous AT methods.
>
> Additionally, several Reviewers (eayv and ZSoG) also recognize the novelty of PBCAT. As noted by Reviewer eayv, one of the strength of this work is that "the method is simple and effective". This is also our stance, "simple" does not necessarily lead to a lack of contribution. In contrast, it enables the scalability and good generalization of our method: It can improve significant robustness against various attacks on several distinct detectors.

---

> ### Author Response · Authors · 2024-11-24
> **Thank you for the valuable review (2/4)**
>
> **W3: Experiments**
>
> 1) Models:  We apologize that we cannot provide meaningful results. According to the experiments of the recent SOTA $l_\infty$-bounded AT [6] work on detectors, the success of $l_\infty$ AT on detectors requires adversarially pre-trained backbones (APB), e.g., an adversarially trained ResNet-50 on the large-scale ImageNet-1K. Our PBCAT also followed this paradigm and used the APB checkpoint provided by Salman et al [7] (Appendix A). Regretfully, the backbone of current YOLO model is a customized network instead of popular networks such as ResNet-50 and we cannot find an APB checkpoint. Performing AT from scratch on the large-scale ImageNet-1K is beyond our recent computational resources. Our preliminary experiments during the rebuttal also showed that the adversarial training of a YOLO-v8 indeed collapsed without using the APB (we only obtained a YOLO-v8 with less than 30 AP$_{50}$ on clean COCO images). But we believe that PBCAT can also improve the robustness of detectors like YOLO-v8 if enough computational resources or the APB are provided, as PBCAT needs no assumption about the structure of the detector and have shown success across diverse detectors, including Faster R-CNN, FOCS, and DN-DETR.
>
> [6] Xiao Li, et al. On the importance of backbone to the adversarial robustness of object detectors. arXiv, 2023.
>
> [7] Salman et al. Do adversarially robust imagenet models transfer better? NeurIPS, 2020
>
>
>
> 2) "Clean" results: We apologize that we cannot provide the comparison under matching the clean AP. Compared to L_inf adversarial training (Li et al., 2023), PBCAT introduces adversarial patches into the images, which makes the adversarial examples more challenging. As a result, the AP on clean images is lowered compared with that of L_inf adversarial training, just like the situation that the clean results of Li et al. (2023) was also lowered compared with the vanilla model (see Table 5 and Table 6). Thus, to defend against the challenging physically realizable attacks, we cannot math the clean AP.
>
> On the other hand, we would like to clarify that the decrease in clean accuracy does not necessarily lead to the robustness gain. For example, in Table 6, Li et al., 2023 (linf-bounded AT) still cannot defend against AdvTexture and AdvCaT even with the sacrifice of clean accuracy. In contrast, PBCAT can significantly improve robustness with slight further sacrifice (0.4 AP)
>
> Finally, we believe the huge improvement on adversarial robustness justify the slight decrease on clean data, especially in many security-critical scenarios. This is also recognized by Reviewer ZSoG: *"*The approach may impact accuracy sometimes, especially when dealing with large datasets like COCO, as shown in Table 5. However, the effectiveness in terms of improved robustness is noteworthy.*"*
>
> 3) Results discussion: Please see our response to W4.
>
> **W4: Presentation-the submission is held back by the writing quality, particularly in the method section, mainly focused around the partially existing formulaic descriptions. For instance, the number of selected sub-patches should be parametrized to better align with the presentation of the ablation study in Section 4.3.2.**
>
> Thank you for your suggestion. The number of selected sub-patches is a hyperparameter, denoted as N. In the patch partitioning and selection process (see Section 3.2), we first divide a square patch into N sub-patches (e.g., 64). Half of these sub-patches are then selected based on the gradients at their respective locations. We have clarified this in the revised version and have proofread the formulations for accuracy.
>
> **W5. Minor comments:**
>
> Thanks for your valuable suggestion. For Algorithm 1, we have modified $m$  to $r$. In addition, we have changed the placement of the tables on Page 9 to make it clearer. Considering that "Clean" performance is not our focus and generally the highest "Clean" model is the Vanilla model without any defense techniques, we follow the style of Li et al., 2023 and do not mark it.

---

> ### Author Response · Authors · 2024-11-24
> **Thank you for the valuable review (3/4)**
>
> *Q1：It is unclear whether the patch selection calculation (i.e., the l2 norm calculations) is performed on the clean image or the adversarial image (the one containing the adversarial patch). 1) Could you please clarify this? 2) Additionally, what is the rationale behind choosing a square-shaped mask? 3) Have you considered experimenting with different norms beyond the l2 norm?*
>
> The patch selection calculation is performed on the gradients of the loss with respect to the clean image. We employed square patches, following the conventions of previous patch-based attacks and adversarial training methods, which typically utilize square patches.
>
> Additionally, PBCAT performs well when the norm calculation is changed from L1 to L2. In response to your suggestion, we trained a new Faster R-CNN with PBCAT that calculates the L1 norm of the gradient and evaluated its accuracy on the Inria dataset. We compared the differences in sub-patch selection between the L1 and L2 norms. Our findings indicate that changing the norm from L2 to L1 does **not** affect the selection of 97.80% of the sub-patches during training. The robustness of the trained model shown below also shows that the choice of different norm calculation does not impact the final robustness much.
>
> ||||
> |-|-|-|
> ||clean(Inria)|AdvPatch|
> |PBCAT (L1）|94.2|68.9|
>
>
> *Q2: In the model training, are the weights initialized to random values or pre-trained weights? If random initialization is used, the object detector may risk overfitting on the Inria dataset, which contains only a few hundred images. This could explain the inconsistencies observed between the results on MS-COCO and Inria.*
>
> Thanks for raising this question. Following Li et al. (2024), we trained PBCAT on the MS-COCO dataset using a pretrained ResNet model as the backbone (see Appendix A). The detectors are not trained on the Inria dataset and it was used only for evaluation (see Section 4.1). Thus, there cannot be overfitting on the Inria dataset. The inconsistency may be attributed to the fact that person detection is relatively simple task (but security-critical) compared with general object detection.
>
> *Q3: In line 345, the total number of sub-patches is set to $n^2=64$ , and in lines 238-239, you mention that the top half are selected, indicating that 32 patches are chosen. However, in the ablation study regarding the number of sub-patches used during the selection process (Table 3), only a single value (16) is presented as a portion of the sub-patches, since using 64 means utilizing the entire set. This leads me to infer that 16 is deemed the optimal value. Does using 32 sub-patches result in better performance? It would be beneficial to explore additional values in this experiment.*
>
> We apologize for the unclear description. Our patch selection strategy involves selecting half of the sub-patches after the patch partitioning. In Table 3, "Sub-patches" refers to the total number of sub-patches created after partitioning, rather than the number of sub-patches selected. Specifically, we divide each patch into either 4×4 = 16, 8×8 = 64, or pixel-level (as shown in Table 3). 64 partition patches (32 selected) are the optimal value we have observed.

---

> ### Author Response · Authors · 2024-11-24
> **Thank you for the valuable review (4/4)**
>
> *Q4: Could you provide some insights into the results presented in Table 2, particularly concerning the "Global" component? I find it challenging to understand why the "Global" component enhances robustness against AdvTexture and AdvCaT, given the significant differences in perturbation styles. Additionally, why does robustness decrease against AdvTexture when the Patch and Partition components are added (Lines 3 and 4)?*
>
> We apologize for the unclear description. As discussed in Section 3.3, texture-based attacks, such as AdvTexture, cover a significant portion of the object, and effectively defending against such attacks ideally requires adversarial training with large-scale, unrestricted perturbations. However, simply increasing the patch size can lead to training instability due to obscuring many of the object's features. To address this issue, we introduced the "Global" linf bounded noise back, which adds global perturbations while constraining the intensity of the perturbations as an alternative to using large patches. Additionally, training with l_infty-bounded global noise has been shown to be effective against various adversarial threats [8, 9], and our subsequent results further confirm these findings.
>
> The partition and selection methods should ideally be used together. As noted in the response to Q5, when using only the partition method we used a **random selection** for sub-patches, which could not select optimal locations and brings randomness into results. For instance, under PGDPatch and AdvCaT, robustness improved, while under AdvPatch and AdvTexture, robustness decreased. But overall, the average robustness using partitioning across the four attacks is higher than that without partitioning, indicating an increasing trend on robustness. Utilizing gradient-based selection to identify optimal locations yields consistent and significant improvements. We have mode it clearer in the revised version.
>
>
>
> [8] Zeyu Wang, et al. Revisiting adversarial training at scale. CVPR, 2024.
>
> [9] Xiao Li, et al. Partimagenet++ dataset: Scaling up part-based models for robust recognition. ECCV, 2024.
>
>
>
> *Q5: Following the above question, it is stated that “Partition” denotes the patch partition strategy. What is the strategy other than “Gradient”? What does Line 4 in Table 2 mean?*
>
> "Partition" refers to the process of dividing a sampled patch into **n × n** sub-patches and retaining half of them; however, it does not specify the selection method. "Gradient" denotes the specific patch selection method that selects sub-patches based on gradient information. In contrast, when "Gradient" is absent, the selection defaults to a random approach. Thus, the experiment in Line 4 of Table 2 utilized "Partition" without "Gradient," meaning that the sampled patch was divided into **n × n** sub-patches, and half were randomly selected. We have clarified this in the revised version.
>
> *Q6: While I acknowledge that the paper focuses on patches attached to objects, it would be beneficial to evaluate the proposed approach against attacks that place patches in different locations (e.g., DPatch) and to study the effect of the "Global" component on these attacks. Demonstrating the ability to mitigate the impact of such patches could significantly enhance the paper's contributions.*
>
> |||
> |-|-|
> ||DPatch|
> |Vanilla|38.8|
> |linf AT|53.1|
> |PBCAT|**56.4**|
>
>
> Thank you for your valuable suggestion. We evaluated the effectiveness of our method under the DPatch attack on the Faster R-CNN model using the Inria dataset. The experimental results demonstrate that the "Global" component does have a positive impact (i.e., linf adversarial training), while PBCAT achieved a higher AP) highlighting the effectiveness of our approach.
>
>
> **If the reviewer agrees with our clarification above and kindly re-evaluates the value of our work, we would be very grateful. We are happy to address any further questions regarding our work.**

---

### Official Review · Reviewer_ZSoG · 2024-10-29

**Soundness:** 3
**Presentation:** 2
**Contribution:** 3
**Rating:** 6
**Confidence:** 4

**Summary:**

The paper proposes a novel adversarial training method designed to defend against various physically realizable attacks on object detection tasks. The perturbation for generating adversarial examples during training includes a global perturbation, constrained by an
ℓ-inf norm with a small budget applied across the entire image, and a local patch, randomly positioned within the bounding box. This local patch is composed of sub-patches, with only some selected to inject a larger budget constraint.

**Strengths:**

- As remarked by different experiments, the proposed method increases the robusteness over different attacks.

- Overall I think that the results are quite intersting, it provides a quite large gap above other strategies.

**Weaknesses:**

- The approach may impact accuracy sometime, especially when dealing with large datasets like COCO, as shown in Table 5. However, the effectiveness in terms of improved robustness is noteworthy.

- The authors could have added metrics on training costs in the table to better clarify possible efficiency with respect to other training strategies

**Questions:**

- The authors mention physically realizable attacks that extend beyond adversarial patches. Why should these represent distinct attacks if they are computed to fool the same model? Adversarial patches could potentially encompass also features of an adversarial t-shirt, as they are capable of generalizing and representing any potential adversarial texture. For instance, at the end of Section 2.3, the authors suggest that real-world adversarial patches may not generalize well to other types of physical attacks, why?

- The adversarial training is applied only for inf-norm bounded attacks. It would be interesting to explore SOTA patch and texture attacks bounded on different norms. What about the robustness against L-2 Attacks? How much it is the model cpaable of etending the robustness against L-2 Attacks?

---

> ### Author Response · Authors · 2024-11-24
> **Thank you for the valuable review**
>
> Thank you for the effort of review. We are happy to see that your think that our work is interesting and provides a quite large gap above previous strategies. **We have uploaded a revised version of our paper, with revisions highlighted in blue for clarity.**  Below we address the detailed comments, and hope that you may find our response satisfactory.
>
> *Q1: The approach may impact accuracy sometime, especially when dealing with large datasets like COCO, as shown in Table 5. However, the effectiveness in terms of improved robustness is noteworthy.*
>
> Thank yor for pointing this out. As discussed in the Limitation section, similar to most AT works [1,2], PBCAT slightly decreased the clean accuracies of detectors on the complex MS-COCO dataset.  It is an open question whether there is an internal trade-off between robustness against physically realizable attacks and clean accuracy. We leave it to be future work. In addition, thanks again for your recognition of effectiveness of PBCAT in improving robustness.
>
>
>
> [1] Hongyang Zhang, et al.Theoretically principled trade-off between robustness and accuracy. ICML, 2019
>
> [2] Xiao Li, et al. On the importance of backbone to the adversarial robustness of object detectors. arXiv preprint arXiv:2305.17438, 2023.
>
>
>
> *Q2: The authors could have added metrics on training costs in the table to better clarify possible efficiency with respect to other training strategies*
>
> Thank you for this valuable suggestion. In our initial submission, we mainly discuss the training cost in in the text. We have added a table in Appendix A of the revised version to better clarify the efficiency compared with linf AT.
>
> *Q3：The authors mention physically realizable attacks that extend beyond adversarial patches. Why should these represent distinct attacks if they are computed to fool the same model? ... For instance, at the end of Section 2.3, the authors suggest that real-world adversarial patches may not generalize well to other types of physical attacks, why?*
>
> We apologize for the unclear description. These methods indeed attack the same model, but they utilize fundamentally different attack approaches. Specifically, adversarial patch attacks craft localized adversarial patterns within a randomly selected fixed region (e.g., a square patch), while adversarial texture attacks craft more pervasive adversarial perturbations that spread across the entire surface of the object, e.g., adversarial modifications to clothing textures that cover most of the surface of an object. **Adversarial texture attacks require 3D modeling of an object** instead of simply putting a patch on the images (patch attack) [3]. Both adversarial patch attacks and adversarial texture attacks are physically realizable attacks. But generally, adversarial texture attacks are more advanced attacks with higher attack success rate [3]. Figure 3 show that the patch attacks and texture-based attacks used in this work are significantly different. We have made it clearer in the revised version of the introduction.
>
>
>
> [3] Physically realizable natural-looking clothing textures evade person detectors via 3d modeling. CVPR, 2023.
>
> *Q4：What about the robustness against L2 Attacks? How much it is the model capable of extending the robustness against L2 Attacks?*
>
> Thank you for your question. We would like to clarify that PBCAT aims to defend against various physically realizable attacks, include patch-based attacks and texture-based attacks. These physically realizable attacks are signifcantly different from conventional $l_p$-bounded attacks. The $l_p$-bounded attacks involve adding a global adversarial perturbation to the images and necessitate manipulation of all image pixels with a $l_p$ budget, which are infeasible in the physical world. And thus the $l_p$-bounded AT using $l_p$-bounded attacks cannot defend against physically realizable attacks well, and vice versa. We have made it clearer in the revised version.
>
>
>
> **If the reviewer agrees with our clarification above, we would be very grateful. We are happy to address any further questions regarding our work.**

---

### Official Review · Reviewer_eayv · 2024-10-31

**Soundness:** 3
**Presentation:** 3
**Contribution:** 3
**Rating:** 6
**Confidence:** 4

**Summary:**

The authors propose a adversarial training method to defend against physically realizable attacks. Specifically, they propose a new adversarial patch attack and use them to train the model.

**Strengths:**

- The method is simple and effective.
- The experimental results and ablation studies are convincing.

**Weaknesses:**

- It is curious that the proposed methods work for naturalistic patch attacks. Experiments on defending naturalistic patch attack will strengthen the paper.
- No black-box experiments are conducted. For example, FastRCNN trained with the proposed method against different datasets and attacks using other surrogate models such as Yolo.
- Hyper-parameter tuning and training time is a concern

**Questions:**

See the Weakness.

---

> ### Author Response · Authors · 2024-11-24
> **Thank you for the valuable review**
>
> Thank you for the effort of review. We are happy to see that your think our work to be simple and effective, with convining results and ablation studies. **We have uploaded a revised version of our paper, with revisions highlighted in blue for clarity.**  Below we address the detailed comments, and hope that you may find our response satisfactory.
>
> *Q1：It is curious that the proposed methods work for naturalistic patch attacks. Experiments on defending naturalistic patch attack will strengthen the paper.*
>
> Thank you for pointing this out. We have conducted relevant experiments, as shown in Table A3 and Appendix D. Specifically, we evaluated PBCAT against Nat-Patch [1], which is one type of naturalistic patch attacks. The experimental results showed that PBCAT can also significantly improve robustness against the naturalistic patch attacks.
>
> *Q2：No black-box experiments are conducted. For example, Faster R-CNN trained with the proposed method against different datasets and attacks using other surrogate models.*
>
> Thank you for this valuable suggestion. We have included the results against several transfer-based black-box attacks in Appendix D.
>
> As per your suggestion, we additionally used the three types of detectors we trained in this work, Faster R-CNN, FCOS, DN-DETR, to perform the black-box transfer attacks. Here we used the AdvPatch attack on the Inria dataset. The adversarial examples generated on the source (surrogate) models (each column) were fed into the target models (each row), and the results are shown below (the results in diagonal represent the white-box attack):
>
> |Source Model|Faster R-CNN|FCOS|DN-DETR|
> |-|-|-|-|
> |Faster R-CNN|77.6|80.7|83.1|
> |FCOS|80.0|58.0|79.3|
> |DN-DETR|69.2|59.9|56.3|
>
>
> We can see that the models trained with our PBCAT can defend black-box attacks using surrogate models even better than white-box attacks.
>
>
>
> *Q3：Hyper-parameter tuning and training time is a concern*
>
> Thanks for your question. We analyzed the detailed effects of the hyper-parameter used in this work in the ablation studies (Table 3 and Table 4). **The results indicate that in a wide range of hyper-parameter selection, PBCAT is effective to imporve robustness against various attacks.** In fact, our two distinct detectors, FCOS and DN-DETR share the same training hyper-parameters and recipes. These results suggest that our method is not highly sensitive to hyperparameter adjustments.
>
> As for the training time, PBCAT maintains the same number of forward and backward passes as standard training, making the training time comparable. As we have discussed in Appendix A, our approach only requires 44 hours for training on 8 NVIDIA 3090 GPUs. In many security-critical scenarios where training time is not the primary concern, we believe PBCAT provides an effective way to enhance robustness.
>
>
>
> [1] Yu-Chih-Tuan Hu, et al. Naturalistic physical adversarial patch for object detectors. ICCV, 2021.
>
>
>
> **If the reviewer agrees with our clarification above and kindly re-evaluates the value of our work, we would be very grateful. We are happy to address any further questions regarding our work.**

---

### Official Review · Reviewer_t6CX · 2024-11-02

**Soundness:** 4
**Presentation:** 4
**Contribution:** 3
**Rating:** 3
**Confidence:** 3

**Summary:**

Early efforts have primarily focused on defending against adversarial patches, leaving adversarial training (AT) against a broader range of physically realizable attacks underexplored. In this work, the authors address this gap by proposing a unified AT method to defend against various physically realizable attacks. They introduce PBCAT, a Patch-Based Composite Adversarial Training strategy, which optimizes the model by combining small-area gradient-guided adversarial patches with imperceptible global adversarial perturbations that cover the entire image. This design enables PBCAT to defend not only against adversarial patches but also against unseen physically realizable attacks, such as adversarial textures. Extensive experiments across multiple settings demonstrate that PBCAT significantly enhances robustness against various physically realizable attacks compared to state-of-the-art defense methods.

**Strengths:**

1.	The topic studied in the paper is practical.
2.	The proposed method demonstrates a degree of generalization, as it does not rely on specific attack algorithms.
3.	The proposed method is effective against common adversarial attack algorithms.
4.	The experiments conducted are relatively comprehensive.

**Weaknesses:**

1. The paper lacks novelty.
2. The authors should emphasize why standard adversarial training cannot effectively address physically realizable attacks and highlight the advantages of the proposed method presented in this paper.
3. In lines 251-253, the authors' findings seem meaningless, as unlimited adversarial noise will inevitably lead to a decline in training performance.
4. Although the training cost of PBCAT is comparable to that of standard training, it still demands additional computational resources due to the gradient post-processing steps (partial partitioning and selection).

**Questions:**

1. What are the differences between square adversarial patches and physically realizable attacks?
2. Why is it necessary to design defense algorithms specifically for these attacks, and what are the limitations of existing defense methods ?
3. What is the purpose of designing a binary mask? Could you please explain?
4. The location of the mask is randomly selected, and then gradient information is used to determine the final patch. What is the difference between this approach and selecting the mask first followed by a random selection of the patch? Is there any advantage to this method ?
5. Why is the adversarial training method presented in this paper inferior to L_\infty-bounded adversarial training when applied to clean data?

---

> ### Author Response · Authors · 2024-11-24
> **Thank you for the valuable review (1/2)**
>
> Thank you for the effort of review. We are encouraged by the appreciation of the practical usefulness, and strong generalization ability of this work. **We have uploaded a revised version of our paper, with revisions highlighted in blue for clarity.**  Below we address the detailed comments, and hope that you may find our response satisfactory.
>
> *Q1: The paper lacks novelty.*
>
> We respectfully disagree this point. As an effective method defending against physically realizable attacks, PBCAT can improve the robustness of different object detectors by a large margin (see Table1). These improvements can be attribute to the novel and well-motivated design  of PBCAT: the patch partitioning and selection strategy, which is distinct from previous patch-based adversarial training methods. Additionally, several reviewers (eayv and ZSoG) also recognize the novelty of PBCAT. Thus, we respectfully disagree with the assertion that our paper lacks novelty.
>
> We believe our work is indeed novel and has several unique contributions compared with previous works, as listed below:
>
> 1. We propose PBCAT, a novel adversarial training method to defend against various physically realizable attacks with a unified model;
> 2. PBCAT closes the gap between adversarial patches and adversarial textures by patch partition and gradient-guided selection techniques;
> 3. Experiments show that PBCAT achieved promising adversarial robustness over diverse
> physically realizable attacks in strong adaptive settings.
>
> *Q2:  The authors should emphasize why standard adversarial training (AT) cannot effectively address physically realizable attacks and highlight the advantages of the proposed method.*
>
> Thanks for your valuable suggestion. We have modified the introduction of our paper to emphasize why standard AT cannot effectively address physically realizable attacks. Generally, standard AT uses human-imperceptible adversarial noises that are bounded by some lp norm for training. This kind of human-imperceptible adversarial noises are significantly different from physically realizable attacks (**see also our response to Q5 and Figure 3**), and thus standard AT cannot address physically realizable attacks well [1,2,3], and usually defending against physically realizable attacks requires AT with adversarial patches, i.e., patch-based AT. PBCAT is a novel patch-based AT method specially designed for physically realizable attacks. Experiments show that PBCAT achieved significant adversarial robustness over standard AT against diverse physically realizable attacks in strong adaptive settings (Table 1).
>
>
>
> [1] Sukrut Rao, et al. Adversarial training against location-optimized adversarial patches. ECCV workshop, 2020.
>
> [2] Tong Wu, et al. Defending against physically realizable attacks on image classification. ICLR, 2020.
>
> [3] Jan Hendrik Metzen, et al. Meta adversarial training against universal patches. ICML, 2021.
>
> *Q3:  The authors' findings seem meaningless, as unlimited adversarial noise will inevitably lead to a decline in training performance.*
>
> We would like to clarify that the adversarial noise used here is not unlimited; it is constrained to specific areas. In fact, the largest patch size used in Table 4 occupies only 16% of the area of a bounding box. In contrast, existing texture-based attacks, such as AdvTexture, have  adversarial noise covering more than 50% of the bounding box area (see Figure 3). Therefore, our findings indicate that we cannot use the adversarial noises generated by texture-based attacks for training directly. As an alternative, we propose to incorporate global imperceptible adversarial perturbations into the patch-based AT. Please see the detailed insight in Section 3.3.
>
> *Q4:  Although the training cost of PBCAT is comparable to that of standard training, it still demands additional computational resources due to the gradient post-processing steps.*
>
> PBCAT maintains the same number of forward and backward passes as standard training, making the training time comparable. **The theoretical computational cost of the additional gradient post-processing is negligible, as it involves only a pooling operation for patch partition and an L2 norm calculation for patch selection.** However, as **we have discussed in Appendix A**, due to our recent implementation, our approach does incur a slight increase in actual computational cost: from 34 hours to 44 hours on 8 NVIDIA 3090 GPUs.
>
> Given the significant improvement in robustness across various physically realizable attacks, e.g., achieving a 29.7% increase in detection accuracy on Faster R-CNN over the state-of-the-art against AdvTexture, we believe that this substantial enhancement justifies the slight increase in actual training cost. This advantage is particularly significant in many security-critical scenarios where training cost is not the primary concern, PBCAT provides an effective way to enhance robustness while maintaining a affordable training expense.

---

> ### Author Response · Authors · 2024-11-24
> **Thank you for the valuable review (2/2)**
>
> *Q5: Differences between square adversarial patches and physically realizable attacks.*
>
> Thanks for raising this question. **Physically realizable attacks** refer to adversarial patterns that can be produced in the physical world and used to fool deep neural networks. Conventional adversarial attacks consider adding human-imperceptible adversarial noises that are bounded by some lp norm, generally recognized as digital adversarial attacks and physically infeasible (we cannot manipulate precise pixels of an image by manipulating the object in the physical world).
>
> Both **patch-based** and texture-based attacks can be implemented in the physical world and thereby they are physically realizable attacks. In Figure 3, we give the visualizations of different physical realizable attacks, including patch-based and  texture-based attacks. Therefore, **square adversarial patch is one type of physically realizable attacks.** We have added a description on these to make it clearer in Section 1 (Introduction).
>
>
>
> *Q6：Why is it necessary to design defense algorithms specifically for these attacks, and what are the limitations of existing defense methods?*
>
> Physically realizable attacks are realistic and severe threats as they can be created in the real physical world. Many works [4,5,6,7] have tried to defend against such attacks. On the other hand, existing defense methods [4,5,6,7] often consider adversarial patches, the simplest form of physically realizable attacks, leaving defense against a wider range of physically realizable attacks under-explored (see Table 1, most existing methods cannot defend against advanced physically realizable attacks such as AdvTexture). Our PBCAT aims to defending against various physically realizable attacks with a unified AT method.
>
>
>
> [4] Muzammal Naseer, et al. Local gradients smoothing: Defense against localized adversarial attacks. WACV, 2019.
>
> [5] Cheng Yu, et al. Defending against universal adversarial patches by clipping feature norms. ICCV, 2021.
>
> [6] Jiang Liu, et al. Segment and complete: Defending object detectors against adversarial patch attacks with robust patch detection. CVPR, 2022.
>
> [7] Ji, N., et al. Adversarial yolo: Defense human detection patch attacks via detecting adversarial patches. arXiv, 2021.
>
>
>
>
>
> *Q7：What is the purpose of designing a binary mask?*
>
> Thanks for your suggestion. The purpose of designing a binary mask is presented in Section 3.2. In short, it is for finding vulnerable areas for effective adversarial training while keeping enough object information (see Figure 1).
>
>
>
> *Q8：The location of the mask is randomly selected, and then gradient information is used to determine the final patch. What is the difference between this approach and selecting the mask first followed by a random selection of the patch? Is there any advantage to this method?*
>
> The primary difference between the two approach lies in the strategy for selecting sub-patches after dividing the patch. Our method selects sub-patches based on gradient magnitudes, while the approach you mentioned uses random selection as an alternative strategy. Generally, the areas with large gradient norms are the vulnerable areas that have a significant impact on the output loss. Thus, using these adversarial noises for training can be more effective.
>
> **We have made a comparison of these two strategies in Table 2**, specifically the fourth row (the method you mentioned) and the fifth row (our strategy). Our strategy can significantly boost the robustness over using random selection.
>
> *Q9：Why is the adversarial training method presented in this paper inferior to $l_\infty$-bounded adversarial training when applied to clean data?*
>
> Compared to $l_\infty$ adversarial training, PBCAT introduces adversarial patches into the images, which makes the adversarial examples more challenging. As a result, the AP on clean images is slightly lowered compared with that of $l_\infty$ adversarial training due to the trade-off between clean accuracy and adversarial robustness [8, 9]. On the other hand, we believe the huge improvement on adversarial robustness justify the slight decrease on clean data. This is also recognized by Reviewer ZSoG: *"*The approach may impact accuracy sometimes, especially when dealing with large datasets like COCO, as shown in Table 5. However, the effectiveness in terms of improved robustness is noteworthy.*"*
>
>
>
> [8] Hongyang Zhang, et al.Theoretically principled trade-off between robustness and accuracy. ICML, 2019
>
> [9] Xiao Li, et al. On the importance of backbone to the adversarial robustness
> of object detectors. arXiv preprint arXiv:2305.17438, 2023.
>
> **If the reviewer agrees with our clarification above and kindly re-evaluates the value of our work, we would be very grateful. We are happy to address any further questions regarding our work.**

---

> > ### Comment · Reviewer_t6CX · 2024-11-27
> >
> > I would like to thank the authors for their patience and thoughtful responses to my questions.
> >
> > After reviewing the authors' replies and considering the feedback from other reviewers, I still believe the paper has some limitations, particularly in terms of methodological innovation. While the authors have made notable efforts to improve the robustness of the detector, the proposed method is relatively simple and does not introduce significant advancements compared to existing adversarial training methods. Despite the authors’ efforts to address the reviewers' concerns and enhance the overall quality of the paper, the lack of substantial innovation has prevented me from revising my initial rating.
> >
> > Thank you again to the authors for their responses.

---

> > > ### Author Response · Authors · 2024-11-27
> > > **Different people have different opinions about "innovation"**
> > >
> > > Thanks for your feedback. Different people have different opinions about "innovation". So we understand and respect your justification of the innovation of our method. But it seems that you agree that our method is simple and effective. For AI research, shouldn't a simple and effective method be welcome by the community? Please note that "simple" doesn't mean lack of innovation; otherwise why didn't previous research figure it out? We AI researchers are trying our best to devise such methods to benefit the community, and we feel unfair if such an effort is discouraged by "lack of innovation".

---

### Author Response · Authors · 2024-11-25
**Hoping for further feedback**

**Dear reviewers,**

We thank you again for the valuable and constructive comments. We are looking forward to hearing from you about any further feedback.

If you find our response satisfactory, we hope you might view this as a sufficient reason to reconsider the rating further.

If you still have questions about our paper, we are willing to answer them and improve our manuscript.

Best, Authors

---

### Author Response · Authors · 2024-12-01

**Dear reviewers,**

We thank you again for the valuable and constructive comments. Considering the deadline on the discussion phase is approaching, we are are eagerly awaiting your further feedback.

If you find our response satisfactory, we hope you might view this as a sufficient reason to reconsider the rating further.

If you still have questions about our paper, we are willing to answer them and improve our manuscript.

Best, Authors

---

### Meta-Review · Area_Chair_XHp1 · 2024-12-20

**Metareview:**

This work introduced a patch-based adversarial training technique to improve object detection models' robustness against patch-based and more recent texture-based attacks. The proposed method involves two types of perturbations: local perturbations applied to the attacked object and a global perturbation affecting the entire image. The global perturbation is aimed at enhancing the robustness against texture-based attacks. The submission compares their technique to one adversarial training (AT) approach and several non-AT methods across three patch-based attacks.  Reviewers agree: (1) the research topic is interesting and important; (2) the proposed method is simple. (3) The experiments are relatively comprehensive. However, there are some key concerns: (1) The motivation of the proposed method is unclear and some key questions are not solved. For example, why are standard AT or other existing methods not available for the task? What are the main and specific challenges of the tasks? (2) The designs of the method are not well explained. For example, what is the purpose of designing a binary mask?  (3) Lack of novelty. A simple, efficient, but effective method is an expected solution for all researchers. However, the main concern is not about the simplicity but the submission did not bring enough insight and novel perspectives to the community. Based on the discussion, we have to reject this version and encourage the resubmission for an improved version.

**Additional Comments On Reviewer Discussion:**

All reviewers provided solid comments in the first round of reviewing. Most of the reviewers join the discussion after the author's rebuttal. Two reviewers (eayv and ZSoG) provide positive scores and two reviewers ( t6CX  and y8aD) tend to reject the paper due to the limited novelty and unclear motivations. After going through the whole comments and discussions, I agree the work should be further enhanced.

---

### Decision · Program_Chairs · 2025-01-22

Reject